# Remanufacturing workshop management in blockchain and digital twin-based cloud remanufacturing service platform

Qin Xiang[1]*, Xugang Zhang[2], Yan Wang[3]

**1** School of Mechanical and Electrical Engineering, Xinjiang Institute of Engineering, Xinjiang, China, **2** Key Laboratory of Metallurgical Equipment and Control Technology, Ministry of Education, Wuhan University of Science and Technology, Wuhan, China, **3** School of Architecture, Technologies and Engineering, University of Brighton, Brighton, United Kingdom

* xiangqin@xjie.edu.cn

## Abstract

The cloud remanufacturing service platform is an online platform that provides various services and resources for the remanufacturing industry. It promotes the digital transformation of the industry, improves resource utilization efficiency, and reduces environmental impact. However, there are prominent issues in sharing remanufacturing data securely, monitoring the remanufacturing process opaquely, and dealing with high uncertainties in remanufacturing operations within the remanufacturing intelligent workshop, which is an important component of the platform. This paper combines blockchain technology and digital twin technology to study the management mode of the remanufacturing workshop under the cloud remanufacturing service platform. Firstly, the overall framework of the cloud remanufacturing service platform is constructed, and a dual-chain structure with expandable subchains for storing transaction data and workshop data is designed. Secondly, by connecting the remanufacturing workshop with smart contracts and digital twin, real-time monitoring, remote collaboration, and data analysis optimization of the remanufacturing workshop are achieved. Digital twin is applied to the remanufacturing process, and a digital twin-based remanufacturing workshop architecture is established. The key technologies involved are analyzed. Finally, the management mode of the automotive remanufacturing workshop under the cloud-based remanufacturing service platform based on blockchain and digital twin is explored.

## Introduction

In the 21st century, the manufacturing industry faces issues such as resource consumption, environmental pollution, and energy depletion. Recycling of waste materials is an inevitable choice for the development of the manufacturing industry, and the development of remanufacturing has economic, resource, and environmental

**Data availability statement:** All files are available from the kaggle database DOI: https://doi.org/10.34740/kaggle/dsv/15036576.

**Funding:** This research is supported by Doctoral Research Start-up Fund(2025XGYBQJ24).

**Competing interests:** The authors have declared that no competing interests exist.

benefits [1]. However, the consumption pattern of intangible services has replaced the consumption pattern of tangible products. If remanufacturing continues to follow traditional methods, it will not meet the new consumer patterns and restrict the development of the remanufacturing industry [2].

Combining the concept of cloud manufacturing [3,4], a cloud remanufacturing service platform is established to centrally manage and operate various remanufacturing resources, forming a dynamically expandable cloud service pool. Users can transparently access online and offline combined services through the network, conveniently utilizing various remanufacturing resources in the process of remanufacturing waste products, enabling centralized resource utilization and decentralized service provision. This can improve the utilization efficiency of remanufacturing resources, reduce costs, create an agile virtual manufacturing environment, promote collaborative cooperation and innovation among different remanufacturing enterprises, and ultimately achieve green manufacturing.

Blockchain, as a distributed database, differs from traditional databases and has characteristics such as decentralization, immutability, traceability, and multi-party maintenance [5–7]. The cloud remanufacturing service platform, as an important part of reverse supply chain transactions, aligns well with the efficient, secure, and trustworthy features of blockchain. This paper combines blockchain technology with the cloud remanufacturing service platform to address issues such as transaction information traceability, transaction credit assurance, user privacy data protection, and remanufacturing workshop monitoring within the platform.

In the cloud remanufacturing service platform, the remanufacturing workshop plays the role of actually executing the remanufacturing activities. It is a physical workspace used for disassembling, cleaning, repairing, testing, evaluating, and assembling waste products to transform them into useful products or materials [8,9]. As part of the cloud remanufacturing service platform, the remanufacturing workshop needs to provide more efficient, traceable, and controllable remanufacturing processes.

Since its introduction, the concept of digital twin has also attracted attention and importance in the industry. Digital twin is a comprehensive set of virtual models that can fully depict potential or real physical manufacturing products from micro to macro perspectives [10–12]. The introduction of the digital twin concept can effectively promote the integration of the physical and information worlds in the remanufacturing workshop, improve the intelligent production level of the workshop, and facilitate the intelligent development of the remanufacturing industry. Through virtual simulation, remote monitoring, decision support, and data sharing, digital twin can enhance the efficiency, quality, and controllability of the remanufacturing process.

In this paper, based on blockchain technology and digital twin technology, a management form for the remanufacturing workshop in the cloud remanufacturing service platform is designed. The data and relevant information of the remanufacturing workshop are stored on the blockchain, ensuring the security and trustworthiness of the data. Moreover, through the integration of smart contracts, the blockchain platform

is connected to the digital twin of the remanufacturing workshop. Smart contracts ensure that the remanufacturing workshop and participants comply with agreements and regulations. Real-time data from the remanufacturing workshop and its virtual model are connected to the cloud remanufacturing service platform. This enables real-time monitoring of the operational status and performance of the remanufacturing workshop, as well as supporting remote collaboration. Participants can remotely observe the status of the remanufacturing workshop through the platform. The digital twin model of the remanufacturing workshop and the data on the blockchain are used for data analysis and optimization.

## Literature review

### Blockchain technology

Blockchain, the technology proposed by Satoshi Nakamoto for Bitcoin, enables mining and transactions by creating an encrypted data structure, providing high security, irreversibility, distribution, transparency, and accuracy [13–15].

Blockchain has been widely applied in the field of cloud manufacturing, similar to cloud remanufacturing. Li et al. [16] proposed a blockchain-based framework to enhance the security and scalability of cloud manufacturing. Yu et al. [17] presented a blockchain-based platform architecture for cloud manufacturing, aiming to improve information transparency and decentralization, where the cloud manufacturing platform exposes manufacturing resources and packages them as services. Aghamohammadzadeh et al. [18] introduced a blockchain-based service composition model platform, which provides an effective collaboration mechanism for service providers in service composition. Due to the continuous growth of service products in cloud manufacturing, Tan et al. [19] proposed a novel service level agreement model that integrates blockchain and smart contracts to address trust issues among cloud service providers, consumers, and third-party monitors. Tong et al. [17] presented a new blockchain-based multi-objective service composition architecture for cloud manufacturing. By combining the cloud remanufacturing service platform with blockchain technology, it is possible to effectively address common issues in manufacturing integration platforms such as trust deficiency, user privacy threats, and untraceable data.

### Digital twin workshop

The concept of digital twin, as one of the key technologies that effectively integrates the physical and virtual worlds of manufacturing, has gained extensive attention and research in recent years [20]. The concept of digital twin was initially proposed by American scholar Grieves in 2003 during a product lifecycle management course [21]. Digital twin creates a digital replica of a physical entity to simulate its operations and behaviors in a real environment, aiming to achieve objectives such as state monitoring, performance optimization, fault diagnosis, and hazard warning [22,23].

Tao et al. [24] first introduced the concept of a digital twin workshop, elucidating its system composition and operational mechanism. They subsequently explored the theories and technologies of information-physical integration in digital twin workshops, providing references for related research and practical applications of digital twin workshops. Bao et al. [25] proposed an ontology-based modeling and evolution method for assembly workshops based on digital twin. Kong et al. [26] presented a data construction method to provide stable and efficient data support for the application of digital twin workshops. Leng et al. [27] established information-physical space connections through distributed digital twin models to create a dynamic autonomous system of various manufacturing resources, facilitating the co-creation of personalized products in intelligent workshops for large-scale customized production. Zhang et al. [28] addressed the challenges of modeling, simulation, and verification in digital twin workshops and proposed a modeling and online training method for digital twin workshops. Liu et al. [29] developed an intelligent scheduling method by leveraging the advantages of digital twin and super networks to generate process plans rapidly and efficiently in workshops. Qian et al. [30] proposed a multi-dimensional data modeling and model validation method for digital twin workshops to tackle issues in data modeling and digital model verification during the construction process.

Digital twin workshops possess characteristics of the integration of virtual and physical realms, data-driven decision-making, and the ability to iterate and optimize throughout the product lifecycle. The introduction of digital twin concepts effectively promotes the integration of the physical and virtual worlds in workshops, enhances the level of intelligent production in workshops, and promotes the development of intelligent manufacturing. Based on these concepts and in conjunction with blockchain technology, this paper designs a management model for remanufacturing workshops under the cloud remanufacturing service platform.

## Blockchain-based cloud remanufacturing service platform

### The framework of blockchain-based remanufacturing service platform

This paper establishes a framework of Blockchain-based Remanufacturing Service Platform, as shown in Fig 1.

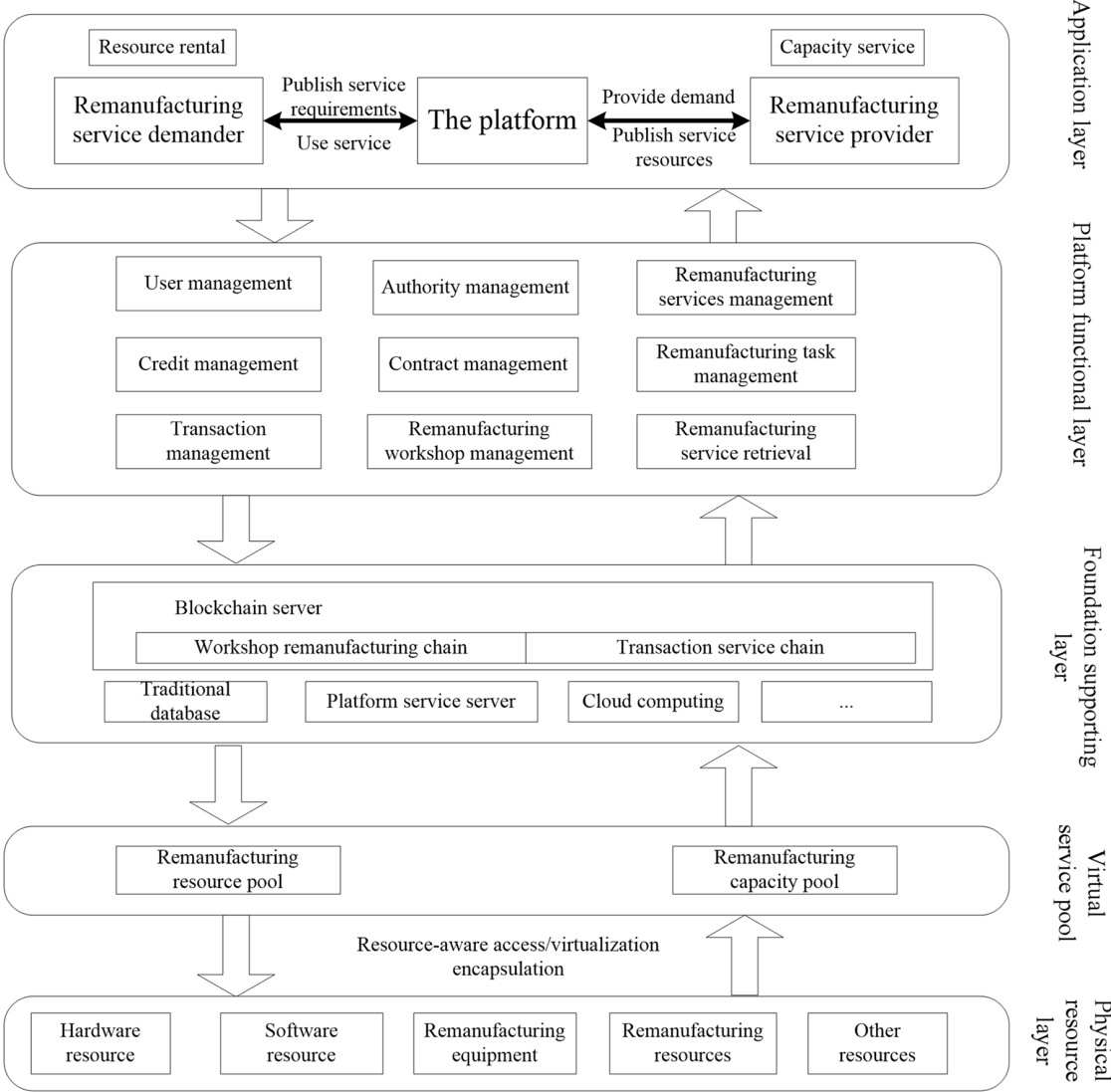

**Fig 1. Framework of blockchain-based remanufacturing service platform.**

(1) Application Layer: This layer caters to three different user identities: remanufacturing service demanders, remanufacturing service providers, and platform operators. Users with different identities utilize different applications within the remanufacturing service platform to fulfill their needs and tasks. Remanufacturing service demanders can access the platform's service publishing, filtering, and matching functionalities to find suitable remanufacturing services for their requirements. Remanufacturing service providers can publish their own remanufacturing services and engage in transactions and collaborations with demanders. Platform operators, on the other hand, manage and monitor the platform's operations and provide relevant support and services.

(2) Platform Function Layer: This layer is responsible for implementing the system's business functions to meet the needs of different business entities. It primarily includes user management (providing functionalities such as user registration, login, and personal information management for remanufacturing service demanders, providers, and platform operators to manage their accounts and personal information), transaction management (including functionalities for transaction publishing, order management, and payment processing to manage the transaction process of remanufacturing services, ensuring security and traceability of transactions), contract management (providing functionalities for creating, managing, and executing smart contracts, allowing remanufacturing service providers and demanders to establish contracts and automatically execute the conditions and operations specified in the contracts), remanufacturing workshop management (encompassing various workshops and related resources involved in the remanufacturing process, including resource scheduling, equipment management, production planning, etc., to ensure smooth progress of remanufacturing tasks), remanufacturing service management (managing the publishing, categorization, and evaluation of remanufacturing services, providing functionalities for service search and filtering so that demanders can find suitable remanufacturing service providers), remanufacturing task management (used for publishing and managing remanufacturing tasks, including task allocation, progress tracking, quality control, etc., to ensure timely and high-quality completion of remanufacturing tasks), and other functionalities.

(3) Foundation Supporting Layer: The foundation supporting layer consists of the relevant technologies and servers that support the operation of the platform, including traditional manufacturing service integration platform components such as traditional databases, platform business servers, cloud computing technologies, and blockchain databases. The design of the infrastructure support layer aims to provide reliable infrastructure and technical support, ensuring the stable operation and data security of the cloud remanufacturing service platform. By employing components such as traditional databases, platform business servers, cloud computing technologies, and blockchain databases, the platform can effectively handle user requests, manage data, and support various functionalities of remanufacturing services.

(4) Virtual Sevice Pool and Physical Resource Layer: By transforming physical resources into virtual services, the cloud remanufacturing service platform establishes a virtual resource pool. This pool encompasses various types of virtual remanufacturing services, including remanufacturing equipment, remanufacturing resources, software resources, hardware resources, etc. Remanufacturing service demanders can select suitable services from the virtual resource pool to achieve flexible configuration and cloud collaboration in the remanufacturing process. The formation of the virtual resource pool provides the cloud remanufacturing service platform with abundant resource choices and efficient service delivery.

## Double chain structure

In the cloud remanufacturing service platform, both supply and demand parties engage in activities such as negotiation and service provider selection while processing order transactions, simultaneously generating transaction service chains. Users upload the enterprise's basic information, remanufacturing task requirements, and available remanufacturing services through the user interface to the transaction service chain. Simultaneously, through the use of digital twin and cloud

computing technologies, the enterprise's remanufacturing resources and capabilities are connected to the internet, and the real-time updates are reflected in the user scheduling information and virtual service pool on the transaction service chain. This allows remanufacturing service providers to provide real-time feedback on their actual capacity and service time arrangements to the virtual service pool and user scheduling information data. The actual task demand time and other related business schedule time data of remanufacturing service demanders are also directly recorded on the chain to ensure the authenticity and reliability of the data. The data used by both parties for remanufacturing service supply-demand matching and business scheduling on the platform are accurate and trustworthy.

After reaching consensus between the transaction parties within the transaction service chain, the workshop remanufacturing chain is generated, and the consensus is transformed into smart contracts within the workshop remanufacturing chain. The remanufacturing workshop then begins organizing remanufacturing-related operations. The main intention is to make sure that the remanufacturing elements, behaviors, and rules strictly conform to the requirements in the actual remanufacturing process to ensure the quality and compliance of remanufacturing. Such arrangements effectively prevent any party from tampering with the execution process, ensuring the reliability and transparency of the remanufacturing process, and providing a fair and trustworthy transaction environment for all parties involved. Through the application of digital technology and smart contracts, the workshop remanufacturing chain can achieve precise control and supervision of the remanufacturing process, improving efficiency and sustainability.

As shown in Fig 2, from the generation timeline of blocks, Remanufacturing service demanders and providers prioritize the generation of transaction service chains based on service reputation, capabilities, and other factors in the stages of service publishing, filtering, and matching. Meanwhile, they push data related to remanufacturing production factors and process control to the workshop remanufacturing chain. Based on the conditions of service consensus, they verify through static and dynamic execution information and provide feedback to the transaction service chain. Furthermore, as the remanufacturing process progresses, new transaction service chains are generated. The smart contract environment is in control of the preparation, execution, and handover of smart contracts during the remanufacturing process. It forms smart contracts for remanufacturing task elements, process controls, and other activities, autonomously supervising the

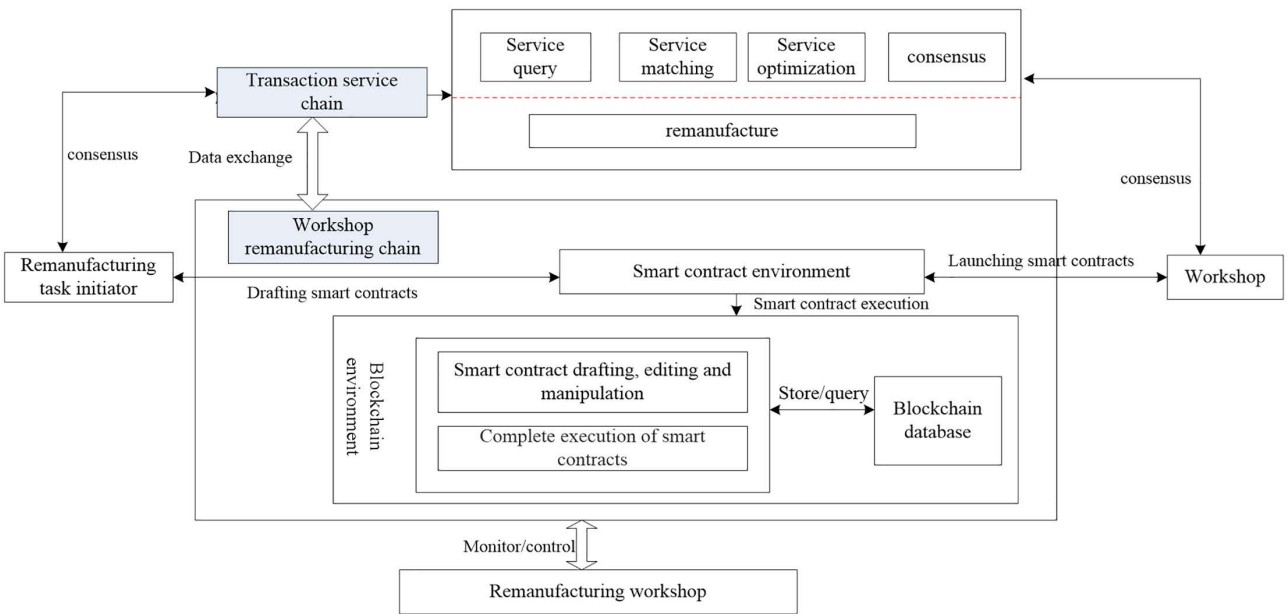

**Fig 2. Interaction model of service transaction chain and workshop remanufacturing chain.**

remanufacturing process and data. It also requires data authentication and rationality from the transaction service chain. In the blockchain environment, operations such as drafting and editing of smart contracts take place. Upon publication, the smart contracts execute automatically in accordance with its predefined logic, encompassing functionalities such as block storage, status queries, and other related operations.

Each node on the blockchain stores a consistent ledger, which leads to significant data redundancy. As the number of nodes and transaction volume increases, it can result in decreased computing speed, lower throughput, and increased latency in the blockchain. As shown in Fig 3, this paper proposes a scalable sub-chain mechanism for the dual-chain blockchain architecture design of the cloud remanufacturing service platform. The main design concept of this mechanism is as follows: the dual-chain blockchain generates sub-chains based on the nature and number of nodes. All nodes on any sub-chain can share the metadata and protocols of that chain. In other words, each node's information is only publicly available within its own sub-chain and is only accessed during transactions. This ensures that nodes can interact with each other while maintaining the privacy of transaction data.

The sub-chains are divided into multiple sub-chains and hosted by different servers when they exceed a certain threshold. The extended sub-chains only allow read and write operations for the nodes on the chain. This scalable sub-chain mechanism enhances the protection of privacy data for stakeholders and balances the overall system load. It provides high computational power and throughput while maintaining low latency.

## Remanufacturing workshop based on digital twin

Remanufacturing production is the process of taking recycled waste products and manufacturing them for resale. It includes production processes such as product recovery, disassembly, cleaning, testing, classification, and reprocessing. The workshop production flow for remanufacturing is illustrated in Fig 4. Due to the nature of remanufacturing, which

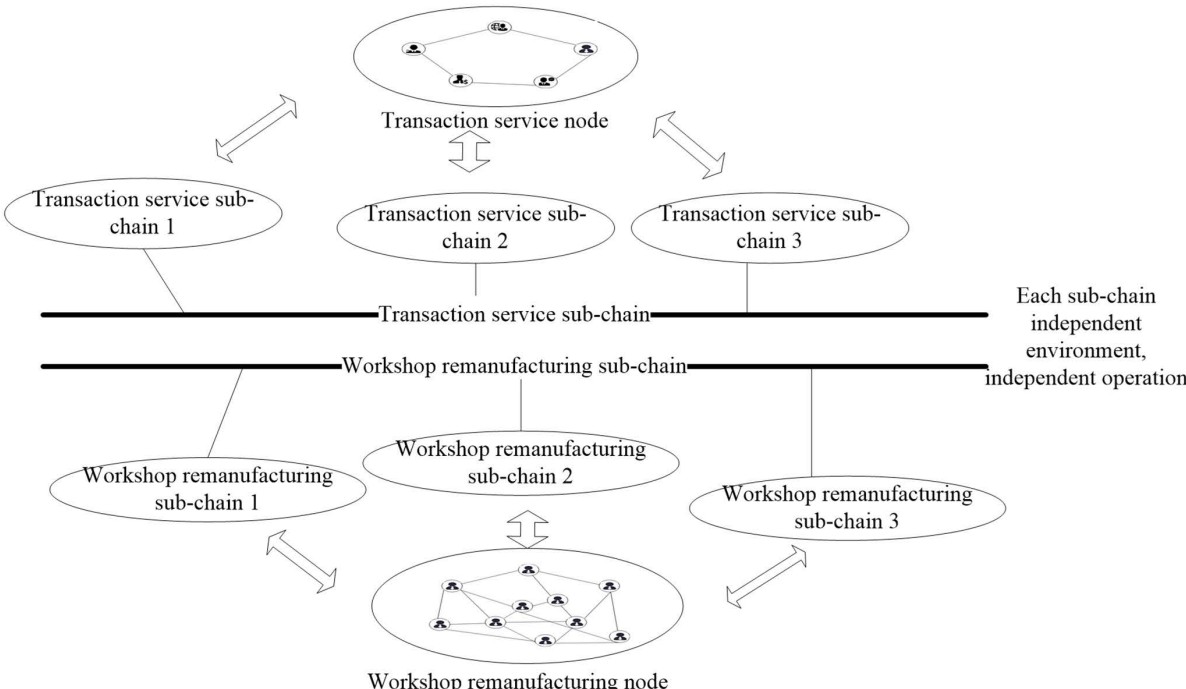

**Fig 3. A double-chain blockchain model that can extend sub-chains.**

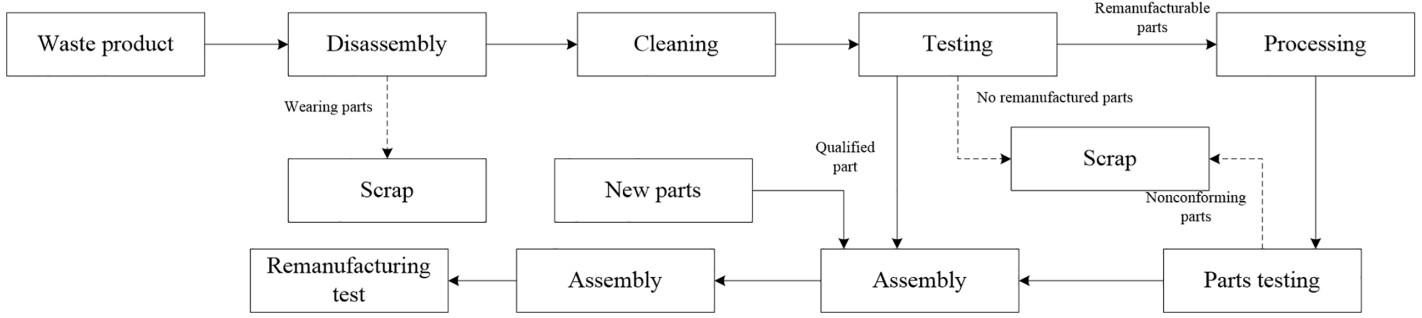

**Fig 4. Remanufacturing process flow chart.**

involves reprocessing discarded products, there are various uncertainties arising from the varying degrees of damage in the recovered products. Remanufacturing also inherits the inherent uncertainties of the original manufacturing process, which continue to affect the remanufacturing process. In addition to the uncertainties that may arise in the original manufacturing process, there are also uncertainties in the stages of waste product recovery, disassembly, testing, and classification.

Based on the remanufacturing production process, a digital twin-based remanufacturing workshop, as shown in the Fig 5, has been designed. Digital twin creates high-fidelity virtual models of physical objects in a digital manner to simulate and predict their behavior, achieving the integration of cyberspace and physical space in remanufacturing. The physical remanufacturing workshop demonstrates the physical processes of the remanufacturing production process, and it executes the remanufacturing tasks of the waste products based on smart contracts. Considering the complexity of the operation process and the uncertainties of waste products, the physical remanufacturing workshop requires strong logistics connectivity, such as the flow of waste products or components between personnel and various equipment, as well as the seamless interaction among the equipment itself. It also needs real-time perception and fusion capabilities of multi-source heterogeneous data to enable collaborative operation of the entire physical operation process. The virtual remanufacturing workshop demonstrates the twinning process. The virtual remanufacturing workshop digitally maps the physical remanufacturing workshop from three aspects: rules, elements, and behaviors, using modeling techniques in the information space. It essentially consists of various models, including operational rules, entity elements, and behavioral features. By simulating and simulating various models in the virtual remanufacturing workshop, the remanufacturing service operation process can be dynamically and accurately evaluated, predicted, and validated. Uncertainty factors can be anticipated, and information related to remanufacturing life and quality issues can be obtained, allowing for timely adjustment and optimization of corresponding process plans.

Digital twin is a digital representation of a physical entity (such as equipment or systems) that is capable of meeting a set of use case requirements. Remanufacturing service twin data mainly includes virtual/physical remanufacturing workshop-related data, remanufacturing service process data, and the fusion of these three types of data. Virtual remanufacturing workshop data primarily consists of the data required for its operation and the data generated during operations, such as various model data (knowledge, rules, etc.), simulation and simulation data, evaluation, analysis, prediction, quality traceability data, etc. Physical remanufacturing workshop data includes physical operation data as well as various operational process specifications, personnel operation experience, and production management standards. Remanufacturing service process data mainly covers the data generated from optimizing and supporting workshop control and prediction in the remanufacturing system, such as data from various information service systems and data for adjusting and optimizing operation plans. These data are stored in traditional database and blockchain database and are further processed to generate derived data through statistical analysis, organization, and other operations.

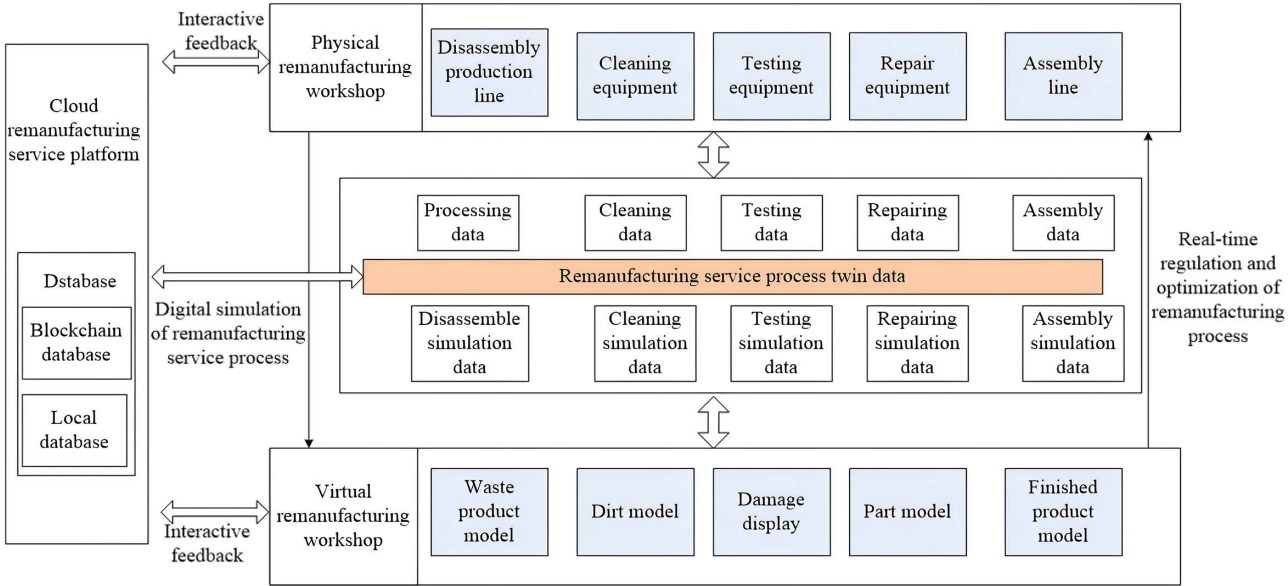

**Fig 5. Digital twin-based remanufacturing workshop.**

The platform accesses both traditional databases and blockchain databases to obtain detailed information related to remanufacturing. This information includes real-time data on sensing device operating status, process parameters, and other important data. While ensuring authentication of processing tasks through blockchain, the platform can flexibly adjust the production schedule of the workshop as needed to achieve an efficient remanufacturing process. The platform closely interacts with the workshop's remanufacturing chain database and smart contract system, using smart contracts to monitor key remanufacturing twin service data. Through this approach, the platform can independently trigger instructions and complete various system tasks according to set rules and logic. The monitoring of smart contracts and independent instruction triggering provide a reliable foundation, enabling the efficient operation of the remanufacturing operation process.

## Design of remanufacturing workshop operation mode

Based on the constructed architecture of a remanufacturing workshop using digital twin and blockchain, starting from the main process of remanufacturing operations, the workshop is divided into five main units: Disassembly, Cleaning, Testing and evaluation, Repair, and Assembly. The following provides a detailed description of the operation process in each unit, leading to an overall description of the operational mode of the remanufacturing workshop.

### Disassembly

Disassembly of end-of-life products is an important prerequisite and critical step in the remanufacturing process, as it determines the maximum utilization of resources for remanufacturing. However, the disassembly process itself involves uncertainties, such as the uncertain quantity of disassembled parts and the uncertain connection methods of components. In the information space, a model of the end-of-life product and the disassembly equipment is constructed. The disassembly management service platform provides services such as displaying technical information/blueprints and controlling the disassembly process, allowing for simulation, emulation, and continuous optimization of the virtual disassembly process. This forms an initial disassembly execution plan.

The virtual/physical disassembly units interact in real-time, with the virtual unit continuously obtaining disassembly-related data from the physical unit and dynamically optimizing the disassembly process. A significant amount of disassembly data is generated during the process and stored in the database, which serves as the driving force for services such as damaged part handling and end-of-life component classification. The digital twin-based disassembly process facilitates the management of uncertainties and enables effective data management, thereby informatizing the disassembly of components. By utilizing digital twin technology, the disassembly process can be better controlled, uncertainties can be managed, and relevant data can be effectively managed. This approach enhances the information processing of component disassembly and supports services such as damaged part handling and end-of-life component classification.

## Cleaning

Cleaning is a relatively straightforward operation in the remanufacturing process, but the quality of cleaning ultimately affects the quality of the remanufactured product. It is necessary to clean the end-of-life products before disassembly and clean the end-of-life components after disassembly, primarily to remove contaminants such as grease, rust, dirt, scale, and carbon deposits from the surfaces of the components. In the digitally-driven cleaning process, digital models are created for the cleaning objects, part contaminants, and cleaning media, and processes such as laser cleaning and ultrasonic cleaning are simulated in the information space, along with corresponding cleaning protocols. Through the interaction between the physical and virtual cleaning spaces, the virtual space obtains data on cleaning force and contamination points from the physical space, and continuously optimizes the cleaning process accordingly. This optimization includes the rational allocation of cleaning materials such as cleaning agents and cleaning equipment. By constructing a digital twin model of the cleaning process, the planning of cleaning materials, cleaning efficiency, and quality can be optimized. The utilization of digital twin technology in the cleaning process enables efficient planning and optimization of cleaning materials, as well as improved cleaning efficiency and quality.

## Testing and evaluation

The testing and evaluation unit in the remanufacturing process primarily focuses on inspecting the cleaned end-of-life components. It evaluates their remaining lifespan, remanufacturability/remanufacturing rate, and predicts the uncertainty of component repairs based on relevant data obtained from the inspections.The testing and evaluation driven by digital twinning first involves constructing a high-fidelity model of the inspected component in the information space. Through the transmission of detection instructions, the component's attribute variables are inspected, including defect testing, geometric measurement, and mechanical property testing (such as hardness, dynamic balance, etc.). Simulation techniques, such as ultrasound, eddy current, and radiography, are employed on computers to detect component damage. The detection data is then controlled and processed to obtain data reflecting the component's state variables, which drive subsequent uncertainty prediction services for component repairs. Simultaneously, the structural analysis of the component model in the information space (e.g., computer-aided engineering (CAE) techniques) is conducted to display the failure mode and component connection methods. This analysis serves as the basis for assessing the component's remaining lifespan, remanufacturability/remanufacturing rate, and provides references for subsequent disassembly and assembly processes.

## Testing and evaluation

The repair of end-of-life components is a key factor in ensuring the quality of remanufactured products. In the remanufacturing process, it is necessary to eliminate variations in the repair process and ensure consistency in the quality performance indicators of the remanufactured components. Based on the damage condition, shape, and structural characteristics of the end-of-life components, as well as the facility layout of the repair unit, the platform provides a proposed repair schedule and repair design plan. The repair design plan is compared and optimized with various schemes in the

case library, taking into account specific repair components, repair time, repair costs, and process routes, resulting in a preliminary repair execution plan. Through real-time interaction between the virtual and physical repair units, the physical unit carries out the repair operations, and based on feedback from the actual operation process, the repair execution plan is iteratively improved and optimized. This interaction between the virtual and physical repair units enables continuous improvement and optimization of the repair execution plan, taking into account the specific repair components, repair time, repair costs, and process routes.

## Assembly

Remanufacturing assembly can generally be seen as the reverse process of disassembly. Due to the numerous uncertainties involved in the disassembly process, assembly is comparatively easier in terms of operations. However, it significantly determines the final quality of the remanufactured product. The remanufactured end product is mainly assembled from three types of components: intact disassembled parts, remanufactured parts, and spare parts. Models of these three types of components are constructed in the information space. Based on the physical components and their connection methods, as well as the reverse process of disassembly, assembly process specifications are formulated. Assembly process simulation is conducted in the information space to develop assembly procedures that guide the physical assembly process. Meanwhile, virtual assembly adjusts and iteratively optimizes the assembly in real-time while obtaining data from the physical assembly. This ensures higher assembly precision and better quality for remanufacturing, as shown in Fig 6.

## Case study

The automotive industry is an important field in the development of the remanufacturing industry. Automotive remanufacturing reduces environmental pollution, increases the reuse of waste resources, and offers significant economic benefits. Automotive remanufacturing services will become a key business project on the cloud remanufacturing service platform. Therefore, this article selects automobiles as a case for analysis and constructs a management model for automotive remanufacturing workshops based on blockchain and digital twin technologies.

The demand for remanufacturing services for automobiles is initiated by the service requester. Smart contracts are generated based on the remanufacturing service requirements. Application services download the remanufacturing files published by the requester through the network service interface and sign the smart contract automatically. They then connect various equipment interfaces to the blockchain system. The automotive remanufacturing process is automatically triggered and executed based on the smart contract signed by both parties. The remanufacturing process twin data is stored in the workshop remanufacturing blockchain database and traditional databases according to the distributed rules of the blockchain. At the same time, users verify their access permissions, view the remanufacturing process data, and monitor the execution of automotive remanufacturing tasks in real-time. Smart contracts require the reception of data that complies with the smart contract specifications in theory when executing the automotive remanufacturing tasks. If data that does not comply with the smart contract specifications is received, an alert is generated, and the workshop manager is notified of the abnormal situation through the user interface. The entire process of executing and verifying automotive remanufacturing tasks is triggered independently based on smart contracts. The blockchain ensures immutability, and any changes to contract parameters by the requester, application services, or other parties will trigger an alert. This series of processes is illustrated in Fig 7.

As shown in Fig 8, the operational process of the automotive remanufacturing workshop under the cloud remanufacturing service platform is explained from the perspective of digital twins. Taking the disassembly unit as an example:

(1) Real-time data sensing and collection: RFID electronic tags are attached to scrapped cars, disassembly equipment, and operators in the disassembly unit. Sensors are also added to the disassembly equipment, making the unit IoT-enabled. Each physical entity can perceive and interact in real-time, and the unit automatically collects sensory data, as well as obtains the automotive remanufacturing disassembly plan and process data.

**Fig 6. Assembly.**

(2) Driving virtual disassembly operations: Through technologies like Mixed Reality (MR), Computer-Aided Design (CAD), Flexsim, DELMLA, and intelligent algorithms, digital twin models of disassembly elements are constructed. The layout optimization of disassembly facilities in the unit is verified. The technical information and data obtained drive virtual disassembly operations of cars through the services provided by the disassembly management system. This includes simulating and optimizing the disassembly process to generate preliminary disassembly plans. The disassembled parts are classified and handled accordingly, and relevant recommendations for processing are provided.

(3) Acquisition of virtual disassembly data: Service data related to disassembly optimization, part handling, and simulation process data are obtained through database interfaces such as ODBC and JDBC. These data are stored and managed in both blockchain and traditional databases. They serve as the driving force for feedback to control and optimize the physical disassembly process.

(4) Physical disassembly control and optimization: For the car disassembly line, the physical disassembly process is controlled and optimized based on the driving force of disassembly twin data. The physical disassembly process follows the optimized disassembly plan obtained through simulation. Disassembly personnel can also directly access disassembly plans and handling suggestions from the information space, providing guidance and assistance during

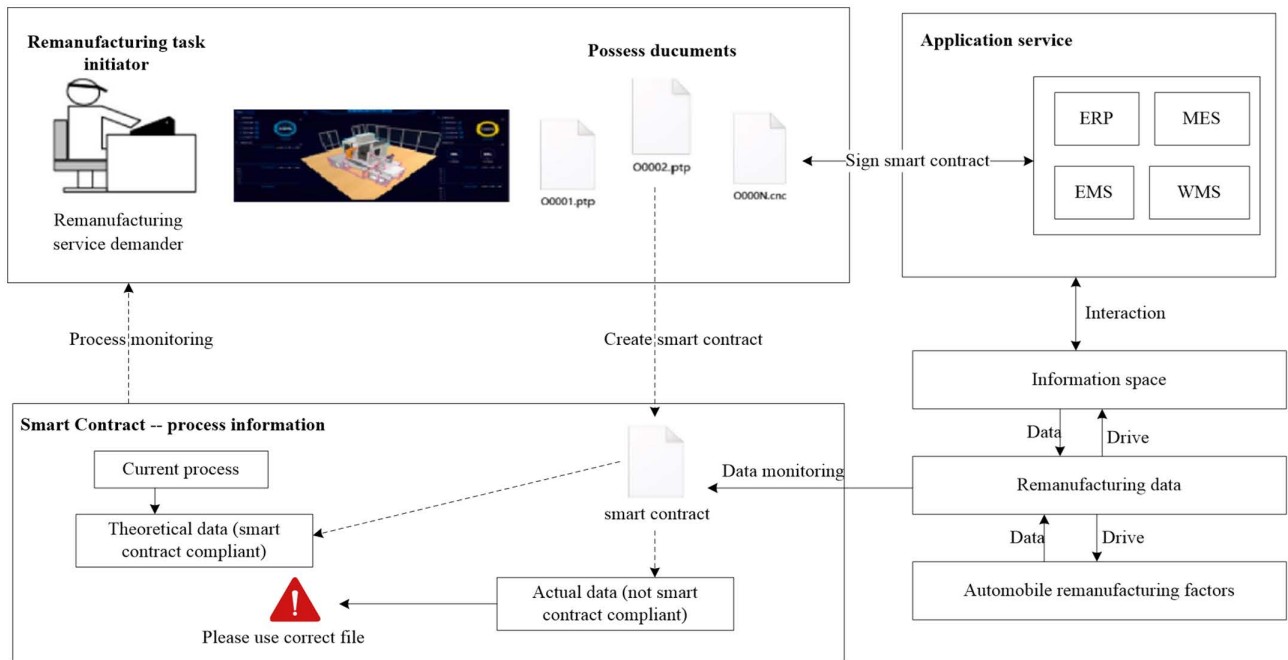

**Fig 7. A blockchain-based automotive remanufacturing workshop.**

the disassembly process. Through the iterative operation of the information flow formed by these four steps, the entire disassembly process can be continuously optimized.

After completing the disassembly of scrapped cars, the same operational process applies to the cleaning unit and subsequent units. Once the automotive remanufacturing workflow is completed, a trial run of the remanufactured car is performed, testing various performance indicators such as power and braking. If any indicators deviate from the remanufacturing parameters and standards, a quality problem analysis and traceability investigation are conducted.

This study proposes a management model for remanufacturing workshops under a cloud remanufacturing service platform, combining blockchain technology and digital twin technology. It aligns with the trends of green, high-quality, efficient, intelligent, and service-oriented development, offering broad applications and development prospects. The specific contributions include:

(1) Researching the overall framework of a cloud remanufacturing service platform based on blockchain. It designs a dual-chain model for storing transaction service data and workshop remanufacturing data, and constructs the architecture of a digital twin remanufacturing workshop under the cloud remanufacturing service platform.

(2) Exploring the implementation approaches of integrating virtual and physical remanufacturing processes through digital twins. The operational process of the remanufacturing workshop is described by dividing it into five operational units.

(3) Exploring the implementation approaches of integrating virtual and physical remanufacturing processes through digital twins. The operational process of the remanufacturing workshop is described by dividing it into five operational units.

(4) Taking automotive remanufacturing as a case study, it demonstrates the feasibility and effectiveness of the proposed management model for remanufacturing workshops under a blockchain and digital twin-based cloud remanufacturing service platform.

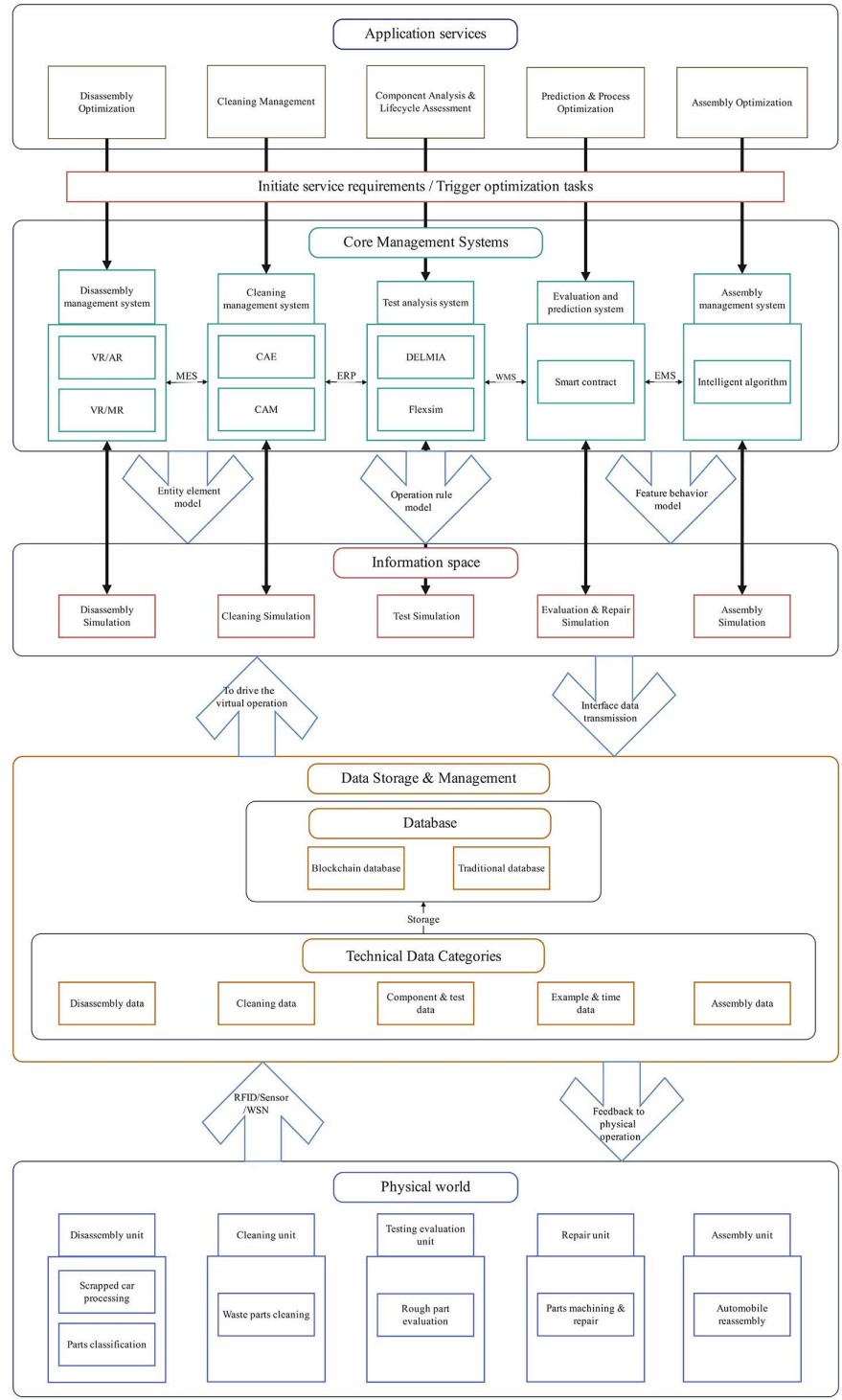

**Fig 8. Automotive remanufacturing shop operations based on digital twins.**

(5) However, this research still has some limitations. For instance, implementing this management model requires certain technological and digital capabilities from remanufacturing companies and stakeholders. Some enterprises with limited resources or relatively outdated technologies may face challenges in technological transformation and equipment upgrades. Additionally, the establishment and maintenance of digital twin models require extensive data collection and processing, while ensuring the accuracy and real-time nature of the models. Some key technological areas are yet to be explored, necessitating further research in relevant aspects.

## Conclusions-future work

Aiming at the core pain points of the workshop under the cloud remanufacturing service platform, including difficult secure data sharing, low process transparency, and high operational uncertainty, this study integrates the decentralized and tamper-proof characteristics of blockchain with the physical-virtual integration advantages of digital twins. A complete management model is constructed, which covers a four-layer architecture, a "double-chain + extensible sub-chain" storage structure, a digital twin workshop system, and an operation mode of five functional units. The feasibility of the proposed model is verified through a case study of automotive remanufacturing. Innovatively, this study proposes a new data storage mode, clarifies the application path of digital twins, and provides a implementable workshop management guideline, thereby offering theoretical and practical support for the digital transformation of the industry. However, there are still unresolved technical limitations, such as the high technical threshold for small and medium-sized enterprises (SMEs), high data collection costs, and difficulties in cross-platform collaboration.

In the future, the research value can be expanded from multiple dimensions. At the technical level, cross-chain architecture and privacy computing will be introduced to enhance data interconnection and security; edge computing and multi-scale modeling will be adopted to improve the real-time performance and accuracy of the twin model; artificial intelligence (AI) will be integrated to realize autonomous decision-making, and metaverse will be combined to address the shortage of technical talents. At the application level, the model will be adapted to the remanufacturing needs of various fields, such as construction machinery, electronic appliances, and aerospace, with targeted optimization of technical schemes. At the implementation level, modular twin tools, light-node alliance chains, and low-cost data collection schemes will be developed to reduce the application threshold for SMEs. At the guarantee level, efforts will be made to establish industry standards such as blockchain data classification for remanufacturing and twin model interfaces, and improve policy subsidy and pilot demonstration mechanisms. Through technical deepening, scenario expansion, threshold reduction, and policy guarantee, this study aims to promote the model from industry pilot to large-scale popularization, and drive the remanufacturing industry towards in-depth transformation of greenization, intelligence, and servitization.

## Author contributions

**Conceptualization:** Qin Xiang, Xugang Zhang.

**Formal analysis:** Qin Xiang, Yan Wang.

**Methodology:** Xugang Zhang.

**Writing – original draft:** Qin Xiang.

**Writing – review & editing:** Yan Wang.

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
