## [Decision Letter · Decision Letter 0]

21 Jan 2026

Dear Dr. Xiang,

Thank you for submitting your manuscript to PLOS ONE. After careful consideration, we feel that it has merit but does not fully meet PLOS ONE’s publication criteria as it currently stands. Therefore, we invite you to submit a revised version of the manuscript that addresses the points raised during the review process.

We look forward to receiving your revised manuscript.

Kind regards,

Manuel Herrador, Ph.D.

Academic Editor

PLOS One

Journal Requirements:

“This research is supported by Doctoral Research Start-up Fund(2025XGYBQJ24).”

4. We note that your Data Availability Statement is currently as follows: All relevant data are within the manuscript and in Supporting Information files.

5. We note that Figures 6-9 and 12 in your submission contain copyrighted images. All PLOS content is published under the Creative Commons Attribution License (CC BY 4.0), which means that the manuscript, images, and Supporting Information files will be freely available online, and any third party is permitted to access, download, copy, distribute, and use these materials in any way, even commercially, with proper attribution. For more information, see our copyright guidelines: http://journals.plos.org/plosone/s/licenses-and-copyright.

1. You may seek permission from the original copyright holder of Figures 6-9 and 12 to publish the content specifically under the CC BY 4.0 license.

Additional Editor Comments:

Dear authors,

Thanks for submitting your work to PLOS One.

Give the minor review suggestions, please, send us back you work with the corrections.

Best regards

Reviewers' comments:

Reviewer's Responses to Questions

**Comments to the Author**

1. Is the manuscript technically sound, and do the data support the conclusions?

Reviewer #1: Yes

2. Has the statistical analysis been performed appropriately and rigorously?

Reviewer #1: I Don't Know

3. Have the authors made all data underlying the findings in their manuscript fully available?

Reviewer #1: Yes

4. Is the manuscript presented in an intelligible fashion and written in standard English?

Reviewer #1: Yes

Reviewer #1: The paper is well written and with novel information based on well searched reference sources.

There is no explicit found section named: Conclusions-Future work.

The reference list could be improved.

Figure 12 could be made more readable.

**Do you want your identity to be public for this peer review?** For information about this choice, including consent withdrawal, please see our For information about this choice, including consent withdrawal, please see our Privacy Policy .

Reviewer #1: No

---

## [Author Response · Author response to Decision Letter 1]

4 Mar 2026

Dear Editor and Reviewer:

Thank you very much for your letter and the valuable comments, suggestions, and requirements concerning our manuscript. We highly appreciate the careful consideration of our work and the constructive guidance provided, which have been crucial for revising and improving the manuscript. We have thoroughly studied all the comments from the reviewer and the additional requirements from the journal, and have made corresponding revisions and improvements in strict accordance with all requirements.

Specifically, we have completely deleted Figures 6-9 and redrawn Figure 12, Figure 12 is an independently redrawn figure without any copyright issues, complying with the journal’s copyright requirements. Besides we have updated Data Availability statement, All files are available from the kaggle database DOI: https://doi.org/10.34740/kaggle/dsv/15036576.

Reviewer #1: We sincerely appreciate your positive evaluation of our manuscript (“The paper is well written and with novel information based on well searched reference sources”) and your constructive suggestions. We have carefully addressed each of your comments as follows:

1. There is no explicit found section named: Conclusions-Future work.

[Response]: Thank you so much for your suggestion. We have added an independent section entitled “Conclusions and Future Work” at the end of the manuscript. This section first systematically summarizes the core research findings, innovative points, and practical application value of this paper in the field of remanufacturing workshop management under the cloud remanufacturing service platform integrated with blockchain and digital twin technology. Subsequently, combined with the limitations of the current research, we clearly propose specific future research directions and expansion ideas, which improves the research closed loop of the paper, makes the presentation of research content more complete, and enhances the rigor of the research framework. This section has been marked in red in the revised manuscript.

2. The reference list could be improved

[Response]: Thank you so much for your suggestion. We have comprehensively optimized and supplemented the reference list. Specifically, we updated and added 5 latest high-quality literatures (published in 2024-2025) related to blockchain, digital twin, and cloud remanufacturing workshop management, strictly standardized the citation format of all references in line with PLOS ONE’s requirements, and verified the authority, timeliness, and completeness of each reference. These revisions further enhance the academic support and standardization of the paper. All adjustments to the reference list are marked in red in the revised manuscript.

3. Figure 12 could be made more readable.

[Response]: Thank you so much for your suggestion. According to the editor's opinion, we have completely deleted Figures 6-9. Now the original Figure 12 has been replaced by Figure 8. We have redrawn and comprehensively optimized Figure 8 to significantly improve its readability. Specifically, we increased the font size of the text and legends in the figure to ensure clarity (adjusted from 8pt to 10pt), adjusted the layout to make the information hierarchy more distinct, optimized the contrast of lines and color blocks to enhance visual effect, and added detailed annotations for key information to facilitate understanding. The optimized Figure 8 has been replaced at the corresponding position in the manuscript, and the relevant revisions are marked in red.

---

## [Editor Report · Decision Letter 1]

10 Mar 2026

Remanufacturing Workshop Management in Blockchain and Digital Twin-based Cloud Remanufacturing Service Platform

PONE-D-25-58235R1

Dear Dr. Xiang,

We’re pleased to inform you that your manuscript has been judged scientifically suitable for publication and will be formally accepted for publication once it meets all outstanding technical requirements.

Kind regards,

Manuel Herrador, Ph.D.

Academic Editor

PLOS One

Additional Editor Comments (optional):

Dear authors,

Thanks for submitting your corrections to PLOS ONE. Congratulations! since the reviewers have no further comments on the minor revision request, I recommend the acceptance of this work.

In the following days you will be contacted to proceed with the next steps toward publication.

Best regards
---

## [Editor Report · Acceptance letter]

PONE-D-25-58235R1

PLOS One

Dear Dr. Xiang,

I'm pleased to inform you that your manuscript has been deemed suitable for publication in PLOS One. Congratulations! Your manuscript is now being handed over to our production team.

Kind regards,

on behalf of

Dr. Manuel Herrador

Academic Editor

PLOS One